# THE ANISOTROPIC NOISE IN STOCHASTIC GRADIENT DESCENT: ITS BEHAVIOR OF ESCAPING FROM MINIMA AND REGULARIZATION EFFECTS

## ABSTRACT

Understanding the behavior of stochastic gradient descent (SGD) in the context of deep neural networks has raised lots of concerns recently. Along this line, we theoretically study a general form of gradient based optimization dynamics with unbiased noise, which unifies SGD and standard Langevin dynamics. Through investigating this general optimization dynamics, we analyze the behavior of SGD on escaping from minima and its regularization effects. A novel indicator is derived to characterize the efficiency of escaping from minima through measuring the alignment of noise covariance and the curvature of loss function. Based on this indicator, two conditions are established to show which type of noise structure is superior to isotropic noise in term of escaping efficiency. We further show that the anisotropic noise in SGD satisfies the two conditions, and thus helps to escape from sharp and poor minima effectively, towards more stable and flat minima that typically generalize well. We verify our understanding through comparing this anisotropic diffusion with full gradient descent plus isotropic diffusion (i.e. Langevin dynamics) and other types of position-dependent noise.

## 1 INTRODUCTION

As a successful learning algorithm, stochastic gradient descent (SGD) was originally adopted for dealing with the computational bottleneck of training neural networks with large-scale datasets (Bottou, 1991). Its empirical efficiency and effectiveness have attracted lots of attention. And thus, SGD and its variants have become standard workhorse for learning deep models. Besides the aspect of empirical efficiency, recently, researchers started to analyze the optimization behaviors of SGD and its impacts on generalization.

The optimization properties of SGD have been studied from various perspectives. The convergence behaviors of SGD for simple one hidden layer neural networks were investigated in (Li & Yuan, 2017; Brutzkus et al., 2017). In non-convex settings, the characterization of how SGD escapes from stationary points, including saddle points and local minima, was analyzed in (Daneshmand et al., 2018; Jin et al., 2017; Hu et al., 2017).

On the other hand, in the context of deep learning, researchers realized that the noise introduced by SGD impacts the generalization, thanks to the research on the phenomenon that training with a large batch could cause a significant drop of test accuracy (Keskar et al., 2017). Particularly, several works attempted to investigate how the magnitude of the noise influences the generalization during the process of SGD optimization, including the batch size and learning rate (Hoffer et al., 2017; Goyal et al., 2017; Chaudhari & Soatto, 2017; Jastrzębski et al., 2017). Another line of research interpreted SGD from a Bayesian perspective. In (Mandt et al., 2017; Chaudhari & Soatto, 2017), SGD was interpreted as performing variational inference, where certain entropic regularization involves to prevent overfitting. And the work (Smith & Le, 2018) tried to provide an understanding based on model evidence. These explanations are compatible with the flat/sharp minima argument (Hochreiter & Schmidhuber, 1997; Keskar et al., 2017), since Bayesian inference tends to targeting the region with large probability mass, corresponding to the flat minima.

However, when analyzing the optimization behavior and regularization effects of SGD, most of existing works only assume the noise covariance of SGD is constant or upper bounded by some

constant, and what role the noise structure of stochastic gradient plays in optimization and generalization was rarely discussed in literature.

In this work, we theoretically study a general form of gradient-based optimization dynamics with unbiased noise, which unifies SGD and standard Langevin dynamics. By investigating this general dynamics, we analyze how the noise structure of SGD influences the escaping behavior from minima and its regularization effects. Several novel theoretical results and empirical justifications are made.

1. We derive a key indicator to characterize the efficiency of escaping from minima through measuring the alignment of noise covariance and the curvature of loss function. Based on this indicator, two conditions are established to show which type of noise structure is superior to isotropic noise in term of escaping efficiency;

2. We further justify that SGD in the context of deep neural networks satisfies these two conditions, and thus provide a plausible explanation why SGD can escape from sharp minima more efficiently, converging to flat minima with a higher probability. Moreover, these flat minima typically generalize well according to various works (Hochreiter & Schmidhuber, 1997; Keskar et al., 2017; Neyshabur et al., 2017; Wu et al., 2017). We also show that Langevin dynamics with well tuned isotropic noise cannot beat SGD, which further confirms the importance of noise structure of SGD;

3. A large number of experiments are designed systematically to justify our understanding on the behavior of the anisotropic diffusion of SGD. We compare SGD with full gradient descent with different types of diffusion noise, including isotropic and position-dependent/independent noise. All these comparisons demonstrate the effectiveness of anisotropic diffusion for good generalization in training deep networks.

The remaining of the paper is organized as follows. In Section 2, we introduce the background of SGD and a general form of optimization dynamics of interest. We then theoretically study the behaviors of escaping from minima in Ornstein-Uhlenbeck process in Section 3, and establish two conditions for characterizing the noise structure that affects the escaping efficiency. In Section 4, we show that the noise of SGD in the context of deep learning meets the two conditions, and thus explains its superior efficiency of escaping from sharp minima over other dynamics with isotropic noise. Various experiments are conducted for verifying our understanding in Section 5, and we conclude the paper in Section 6.

## 2  BACKGROUND

In general, supervised learning usually involves an optimization process of minimizing an empirical loss over training data, $L(\theta) := 1/N \sum_{i=1}^{N} \ell(f(x_i; \theta), y_i)$, where $\{(x_i, y_i)\}_{i=1}^{N}$ denotes the training set with $N$ *i.i.d.* samples, the prediction function $f$ is often parameterized by $\theta \in \mathbb{R}^D$, such as deep neural networks. And $\ell(\cdot, \cdot)$ is the loss function, such as mean squared error and cross entropy, typically corresponding to certain negative log likelihood. Due to the over parameterization and non-convexity of the loss function in deep networks, there exist multiple global minima, exhibiting diverse generalization performance. We call those solutions generalizing well good solutions or minima, and vice versa.

**Gradient descent and its stochastic variants** A typical approach to minimize the loss function is gradient descent (GD), the dynamics of which in each iteration $t$ is, $\theta_{t+1} = \theta_t - \eta_t g_0(\theta_t)$, where $g_0(\theta_t) = \nabla_\theta L(\theta_t)$ denotes the full gradient and $\eta_t$ denotes the learning rate. In non-convex optimization, a more useful kind of gradient based optimizers act like GD with an unbiased noise, including gradient Langevin dynamics (GLD), $\theta_{t+1} = \theta_t - \eta_t g_0(\theta_t) + \sigma_t \epsilon_t, \epsilon_t \sim \mathcal{N}(0, I)$, and stochastic gradient descent (SGD), during each iteration $t$ of which, a minibatch of training samples with size $m$ are randomly selected, with index set $B_t \subset \{1, 2, \ldots, N\}$, and a stochastic gradient is evaluated based on the chosen minibatch, $\tilde{g}(\theta_t) = \sum_{i \in B_t} \nabla_\theta \ell(f(x_i; \theta_t), y_i)/m$, which is an unbiased estimator of the full gradient $g_0(\theta_t)$. Then, the parameters are updated with some learning rate $\eta_t$ as $\theta_{t+1} = \theta_t - \eta_t \tilde{g}(\theta_t)$. Denote $g(\theta) = \nabla_\theta \ell((f(x; \theta), y)$, the gradient for loss with a single data point $(x, y)$, and assume that the size of minibatch is large enough for the central limit theorem to

hold, and thus $\tilde{g}(\theta_t)$ follows a Gaussian distribution (Mandt et al., 2017; Li et al., 2017),

$$\tilde{g}(\theta_t) \sim \mathcal{N}\left(g_0(\theta_t), \frac{1}{m}\Sigma(\theta_t)\right), \text{ where } \Sigma(\theta_t) \approx \frac{1}{N}\sum_{i=1}^{N}\left(g(\theta_t; x_i) - g_0(\theta_t)\right)\left(g(\theta_t; x_i) - g_0(\theta_t)\right)^T. \tag{1}$$

Note that the covariance matrix $\Sigma$ depends on the model architecture, dataset and the current parameter $\theta_t$. Now we can rewrite the update of SGD as,

$$\theta_{t+1} = \theta_t - \eta_t g_0(\theta_t) + \frac{\eta_t}{\sqrt{m}}\epsilon_t, \quad \epsilon_t \sim \mathcal{N}\left(0, \Sigma(\theta_t)\right). \tag{2}$$

Inspired by GLD and SGD, we may consider a general kind of optimization dynamics, namely, *gradient descent with unbiased noise*,

$$\theta_{t+1} = \theta_t - \eta_t g_0(\theta_t) + \sigma_t \epsilon_t, \quad \epsilon_t \sim \mathcal{N}\left(0, \Sigma_t\right). \tag{3}$$

For small enough constant learning rate $\eta_t = \eta$, the above iteration in Eq. (3) can be treated as the numerical discretization of the following stochastic differential equation (Li et al., 2017; Jastrzębski et al., 2017; Chaudhari & Soatto, 2017),

$$d\theta_t = -\nabla_\theta L(\theta_t)\,dt + \sqrt{\eta\sigma_t^2\Sigma_t}\,dW_t. \tag{4}$$

Considering $\sqrt{\eta\sigma_t^2\Sigma_t}$ as the coefficient of noise term, existing works (Hoffer et al., 2017; Jastrzębski et al., 2017) studied the influence of noise magnitude of SGD on generalization, i.e. $\eta\sigma_t^2 = \eta/m$.

In this work, we focus on studying the benefits of anisotropic structure of $\Sigma_t$ in SGD helping escape from minima by bridging the covariance matrix with the Hessian of the loss surface, and its implicit regularization effects on generalization, especially in deep learning context. For the purpose of eliminating the influence of the noise magnitude, we constrain it to be a constant when studying different structures of noise covariance. The noise magnitude could be evaluated as the expectation of the squared norm of the noise vector,

$$\mathbb{E}[(\sqrt{\eta}\sigma_t\epsilon_t)^T(\sqrt{\eta}\sigma_t\epsilon_t)] = \eta\sigma_t^2\mathbb{E}[\epsilon^T\epsilon] = \eta\sigma_t^2\,\mathrm{Tr}\,\mathbb{E}[\epsilon\epsilon^T] = \eta\sigma_t^2\,\mathrm{Tr}\,\Sigma_t. \tag{5}$$

Thus, we introduce the following constraint,

$$\text{given time } t, \quad \eta\sigma_t^2\,\mathbf{Tr}\,(\Sigma_t) \text{ is constant}. \tag{6}$$

From the statistical physics point of view, $\mathrm{Tr}(\eta\sigma_t^2\Sigma_t)$ characterizes the kinetic energy (Gardiner), thus it is natural to force the energy to be unchanging, otherwise it is trivial that the higher the energy is, the less stable the system is.

For simplicity, we absorb $\eta\sigma_t^2$ into $\Sigma_t$, denoting $\eta\sigma_t^2\Sigma_t$ as $\Sigma_t$. If not pointed out, the subscript $t$ of matrix $\Sigma_t$ is omitted to emphasize that we are fixing $t$ and discussing the varying structure of $\Sigma$.

## 3 THE BEHAVIORS OF ESCAPING FROM MINIMA IN ORNSTEIN-UHLENBECK PROCESS

For a general loss function $L(\theta) = \mathbb{E}_X \ell_X(\theta)$ (the expectation could be either population or empirical), where $X$ denotes data example and $\theta$ denoted parameters to be optimized, under suitable smoothness assumptions, the SDE associated with the gradient variant optimizer as shown in Eq. (4) can be written as follows (Li et al., 2017; Jastrzębski et al., 2017; Chaudhari & Soatto, 2017; Hu et al., 2017), with little abuse of notation,

$$d\theta_t = -\nabla_\theta L(\theta_t)\,dt + \Sigma_t^{\frac{1}{2}}\,dW_t. \tag{7}$$

Let $L_0 = L(\theta_0)$ be one of the minimal values of $L(\theta)$, then for a fixed $t$ small enough (such that $L_t - L_0 \geq 0$), $\mathbb{E}_{\theta_t}[L_t - L_0]$ characterizes the efficiency of $\theta$ escaping from the minimum $\theta_0$ of $L(\theta)$. It is natural to measure the escaping efficiency using $\mathbb{E}[L_t - L_0]$ since it characterizes the increase of the potential, i.e., the increase of the loss $L$. And also note that $L_t - L_0 \geq 0$, for any $\delta > 0$, the escaping probability $P(L_t - L_0 \geq \delta)$ can be controlled by the expectation $\mathbb{E}[L_t - L_0]$ since by Markov's inequality, we have $P(L_t - L_0 \geq \delta) \leq \frac{\mathbb{E}[L_t - L_0]}{\delta}$.

**Proposition 1** (Escaping efficiency for general process). *For the process (7), provided mild smoothness assumptions, the escaping efficiency from the minimum $\theta_0$ is,*

$$\mathbb{E}[L_t - L_0] = -\int_0^t \mathbb{E}\left[\nabla L^T \nabla L\right] + \int_0^t \frac{1}{2}\mathbb{E}\operatorname{Tr}(H_t \Sigma_t)\,\mathrm{d}t, \tag{8}$$

*where $H_t$ denotes the Hessian of $L(\theta_t)$ at $\theta_t$.*

We provide the proof in Appendix, and the same for the other propositions.

The escaping efficiency for general processes is hard to analyze due to the intractableness of the integral in Eq. (8). However, we may consider the second-order approximation locally near the minima $\theta_0$, where $L(\theta) \approx L_0 + \frac{1}{2}(\theta - \theta_0)^T H(\theta - \theta_0)$. Without losing generality, we suppose $\theta_0 = 0$. Further, suppose that $H$ is a positive definite matrix and the diffusion covariance $\Sigma_t = \Sigma$ is constant for $t$. Then the SDE (7) becomes an Ornstein-Uhlenbeck process,

$$\mathrm{d}\theta_t = -H\theta_t\,\mathrm{d}t + \Sigma^{\frac{1}{2}}\,\mathrm{d}W_t, \quad \theta_0 = 0. \tag{9}$$

**Proposition 2** (Escaping efficiency of Ornstein-Uhlenbeck process). *For Ornstein-Uhlenbeck process (9), with $t$ small enough, the escaping efficiency from minimum $\theta_0 = 0$ is,*

$$\mathbb{E}[L_t - L_0] = \frac{1}{4}\operatorname{Tr}\left(\left(I - e^{-2Ht}\right)\Sigma\right) \approx \frac{t}{2}\operatorname{Tr}\left(H\Sigma\right). \tag{10}$$

Inspired by Proposition 1 and Proposition 2, we propose $\mathbf{Tr}\,(\boldsymbol{H\Sigma})$ as an empirical indicator measuring the efficiency for a stochastic process escaping from minima. Now we turn to analysis which kind of noise covariance structure $\Sigma$ will benefit escaping sharp minima, under the constraint Eq. (6).

Firstly, for the isotropic loss surface, i.e., $H = \lambda I$, the escaping efficiency is $\mathbb{E}[L_t - L_0] = \frac{\lambda t}{2}\operatorname{Tr}\Sigma$, which is invariant under the constraint that $\operatorname{Tr}\Sigma$ is constant (Eq. (6)). Thus it is only nontrivial to study the impact of noise structure when the Hessian of loss surface is anisotropic.

Secondly, $H$ and $\Sigma$ being semi-positive definite, to achieve the maximum of $\operatorname{Tr}(H\Sigma)$ under constraint (6), $\Sigma$ should be $\Sigma^* = (\operatorname{Tr}\Sigma) \cdot \lambda_1 u_1 u_1^T$, where $\lambda_1, u_1$ are the maximal eigenvalue and corresponding unit eigenvector of $H$. Note that the rank-1 matrix $\Sigma^*$ is highly anisotropic. More generally, the following Proposition 3 characterizes one kind of anisotropic noise significantly outperforming isotropic noise in order of number of parameters $D$, given $H$ is ill-conditioned.

**Proposition 3** (The benefits of anisotropic noise). *With semi-positive definite $H$ and $\Sigma$, assume*

*(1) $H$ is ill-conditioned. Let $\lambda_1 \geq \lambda_2 \geq \ldots, \geq \lambda_D \geq 0$ be the eigenvalues of $H$ in descent order, and for some constant $k \ll D$ and $d > \frac{1}{2}$,*

$$\lambda_1 > 0, \qquad \lambda_{k+1}, \lambda_{k+2}, \ldots, \lambda_D < \lambda_1 D^{-d}, \tag{11}$$

*(2) $\Sigma$ is "aligned" with $H$. Let $u_i$ be the corresponding unit eigenvector of eigenvalue $\lambda_i$, for some projection coefficient $a > 0$,*

$$u_1^T \Sigma u_1 \geq a\lambda_1 \frac{\operatorname{Tr}\Sigma}{\operatorname{Tr}H}, \tag{12}$$

*then we have the benefit of the anisotropic noise over the isotropic one in term of escaping efficiency, which can be characterized by the follow ratio,*

$$\frac{\operatorname{Tr}\left(H\Sigma\right)}{\operatorname{Tr}(H\bar{\Sigma})} = \mathcal{O}\left(aD^{(2d-1)}\right), \tag{13}$$

*where $\bar{\Sigma} = \frac{\operatorname{Tr}\Sigma}{D}I$ denotes the covariance of isotropic noise, to meet the constraint Eq. (6).*

To give some geometric intuitions on the left hand side of Eq. (12), let the maximal eigenvalue and its corresponding unit eigenvector of $\Sigma$ be $\gamma_1, v_1$, then the right hand side has a lower bound as $u_1^T \Sigma u_1 \geq u_1^T v_1 \gamma_1 v_1^T u_1 = \gamma_1 \langle u_1, v_1 \rangle^2$. Thus if the maximal eigenvalues of $H$ and $\Sigma$ are aligned in proportion, $\gamma_1 / \operatorname{Tr}\Sigma \geq a_1 \lambda_1 / \operatorname{Tr}H$, and the angle of their corresponding unit eigenvectors is close to zero, $\langle u_1, v_1 \rangle \geq a_2$, the second condition Eq. (12) in Proposition 3 holds for $a = a_1 a_2$.

Typically, in the scenario of modern deep neural networks, due to the over-parameterization, Hessian and the gradient covariance are usually ill-conditioned and anistropic near minima, as shown by (Sagun et al., 2017) and (Chaudhari & Soatto, 2017). Thus the first condition in Eq. (11) usually holds for deep neural networks, and we further justify it by experiments in Section 5.3. Therefore, in the following section, we turn to focus on how the gradient covariance, i.e. the covariance of SGD noise meets the second condition of Proposition 3 in the context of deep neural networks.

## 4 THE ANISOTROPIC NOISE OF SGD IN DEEP NETWORKS

In this section, we mainly investigate the anisotropic structure of gradient covariance in SGD, and explore its connection with the Hessian of loss surface.

**Around the true parameter**  According to the classic statistical theory (Pawitan, 2001, Chap. 8), for population loss $L(\theta) = \mathbb{E}_X \ell(\theta)$, with $\ell$ being the negative log likelihood, when evaluating at the true parameter $\theta^*$, there is the exact equivalence between the Hessian $H$ of the population loss and *Fisher information matrix $F$*,

$$F(\theta^*) := \mathbb{E}_X[\nabla_\theta \ell(\theta^*) \nabla_\theta \ell(\theta^*)^T] = \mathbb{E}_X[\nabla_\theta^2 \ell(\theta^*)] = \nabla_\theta^2 L(\theta^*) =: H(\theta^*). \tag{14}$$

In practice, with the assumptions that the sample size $N$ is large enough (i.e. indicating asymptotic behavior) and suitable smoothness conditions, when the current parameter $\theta_t$ is not far from the ground truth, Fisher is close to Hessian. Thus we can obtain the following approximate equality between gradient covariance and Hessian,

$$\Sigma(\theta_t) = F(\theta_t) - \nabla_\theta L^T(\theta_t) \nabla_\theta L(\theta_t) \approx F(\theta_t) \approx H(\theta_t).$$

The first approximation is due to the dominance of noise over the mean of gradient in the later stage of SGD optimization, which has been shown in (Shwartz-Ziv & Tishby, 2017). A similar experiment as (Shwartz-Ziv & Tishby, 2017) has been conducted to demonstrate this observation, which is left in Appendix due to the limit of space.

In the following, we theoretically characterize the closeness between $\Sigma$ and $H$ in the context of one hidden layer neural networks; and show that the gradient covariance introduced by SGD indeed has more benefits than isotropic one in term of escaping from minima, provided some assumptions.

**One hidden layer neural network with fixed output layer parameters**  For binary classification neural network with one hidden layer in classic setups (with softmax and cross-entropy loss), we have following results to globally bound Fisher and Hessian with each other.

**Proposition 4** (The relationship between Fisher and Hessian in one hidden layer neural network).
*Consider the binary classification problem with data $\{(x_i, y_i)\}_{i\in I}, y \in \{0, 1\}$, and typical (either population or empirical) loss as $L(\theta) = \mathbb{E}[\phi \circ f(x; \theta)]$, where $f$ denotes the output of neural network, and $\phi$ denotes the cross-entropy loss with softmax,*

$$\phi(f(x), y) = -\left( y \log \frac{e^{f(x)}}{1 + e^{f(x)}} + (1 - y) \log \frac{1}{1 + e^{f(x)}} \right), y \in \{0, 1\}.$$

*If: (1) the neural network $f$ is with one hidden layer and piece-wise linear activation. And the parameters of output layer are fixed during training; (2) the optimization happens on a set $U$ such that, $f(x; \theta) \in (-C, C), \forall \theta \in U, \forall x$, i.e., the output of the classifier is bounded during optimization.*

*Then, we have the following relationship between (either population or empirical) Fisher $F$ and Hessian $H$ almost everywhere:*

$$e^{-C} F(\theta) \preceq H(\theta) \preceq e^C F(\theta).$$

*$A \preceq B$ means that $(B - A)$ is semi-positive definite.*

There are a few remarks on Proposition 4. Firstly, as shown in (Brutzkus et al., 2017), the considered neural networks in Proposition 4 are non-convex and have multiple minima, and thus it is still nontrivial to consider the escaping from minima. Secondly, the Proposition 4 holds in both population and empirical sense, since the proof does not distinguish the two circumstances. Thirdly, the bound

between $F$ and $H$ holds "globally" in the set $U$ where the output $f$ is bounded, rather than merely around the true global minima as discussed previously.

By Proposition 4, the following relationship between gradient covariance and Hessian could be derived.

**Proposition 5** (The relationship between gradient covariance and Hessian in one hidden layer neural network). *Assume the conditions in Proposition 4 hold, then for some small $\delta > 0$ and for $\theta$ close enough to minima $\theta^*$ (local or global),*

$$u^T \Sigma u \geq e^{-2(C+\delta)} \lambda \frac{\operatorname{Tr} \Sigma}{\operatorname{Tr} H} \tag{15}$$

*holds for any positive eigenvalue $\lambda$ and its corresponding unit eigenvector $u$ of Hessian $H$.*

As a direct corollary of Proposition 5, for such neural networks, the second condition Eq. (12) in Proposition 3 holds in a very loose sense.

Therefore, based on the discussion on population loss around the true parameters and one hidden layer neural network with fixed output layer parameters, given the ill-conditioning of $H$ due to the over-parameterization of modern deep networks, according to Proposition 3, we can conclude the noise structure of SGD helps to escape from sharp minima much faster than the dynamics with isotropic noise, and converge to flatter solutions with a high probability. These flat minima typically generalize well (Hochreiter & Schmidhuber, 1997; Keskar et al., 2017; Neyshabur et al., 2017; Wu et al., 2017). Thus, we attribute such properties of SGD on its better generalization performance comparing to GD, GLD and other dynamics with isotropic noise (Hoffer et al., 2017; Goyal et al., 2017; Keskar et al., 2017).

In the following, we conduct a series of experiments systematically to verify our understanding on the behavior of escaping from minima and its regularization effects for different optimization dynamics.

## 5 EXPERIMENTS

To better understanding the behavior of anisotropic noise different from isotropic ones, we introduce dynamics with different kinds of noise structure to empirical study with, as shown on Table 1.

Table 1: Compared dynamics defined in Eq. (3). For **GLD dynamic**, **GLD diagonal**, **GLD Hessian** and **GLD 1st eigvec**($H$), $\sigma_t$ are adjusted to make $\sigma_t \epsilon_t$ share the same expected norm as that of SGD. For **GLD leading**, $\sigma_t$ is same as in SGD. Note that **GLD 1st eigvec**($H$) achieves the best escaping efficiency as our indicator suggested.

| | Noise $\epsilon_t$ | Remarks |
|---|---|---|
| **SGD** | $\epsilon_t \sim \mathcal{N}\left(0, \Sigma_t^{\text{sgd}}\right)$ | $\Sigma_t^{\text{sgd}}$ is defined as in Eq. (1), and $\sigma_t = \frac{\eta_t}{\sqrt{m}}$ |
| **GLD constant** | $\epsilon_t \sim \mathcal{N}\left(0, I\right)$ | $\sigma_t$ is a tunable constant |
| **GLD dynamic** | $\epsilon_t \sim \mathcal{N}\left(0, I\right)$ | $\sigma_t$ is adjusted to make $\sigma_t \epsilon_t$ share the same expected norm as that of SGD |
| **GLD diagonal** | $\epsilon_t \sim \mathcal{N}\left(0, \operatorname{diag}(\Sigma_t)\right)$ | The covariance $\operatorname{diag}(\Sigma_t)$ is the diagonal of the covariance of SGD noise. |
| **GLD leading** | $\epsilon_t \sim \mathcal{N}\left(0, \tilde{\Sigma}_t\right)$ | $\tilde{\Sigma}_t = \sum_{i=1}^{k} \gamma_i v_i v_i^T$. $\gamma_i, v_i$ are the first $k$ leading eigenvalues and corresponding eigenvalues of the covariance of SGD noise, respectively. (A low rank approximation of $\Sigma_t^{\text{sgd}}$) |
| **GLD Hessian** | $\epsilon_t \sim \mathcal{N}\left(0, \tilde{H}_t\right)$ | $\tilde{H}_t$ is a low rank approximation of the Hessian matrix of loss $L(\theta)$ by its the first $k$ leading eigenvalues and corresponding eigenvalues. |
| **GLD 1st eigven**($H$) | $\epsilon_t \sim \mathcal{N}\left(0, \lambda_1 u_1 u_1^T\right)$ | $\lambda_1, u_1$ are the maximal eigenvalue and its corresponding unit eigenvector of the Hessian matrix of loss $L(\theta_t)$. |

### 5.1 TWO-DIMENSIONAL TOY EXAMPLE

We design a 2-D toy example $L(w_1, w_2)$ with two basins, a small one and a large one, corresponding to a sharp and flat minima, $(1, 1)$ and $(-1, -1)$, respectively, both of which are global minima.

Please refer to Appendix for the detailed constructions. We initialize the dynamics of interest with the sharp minimum $(w_1, w_2) = (1, 1)$, and run them to study their behaviors escaping from this sharp minimum.

To explicitly control the noise magnitude, we only conduct experiments on GD, GLD const, GLD diag, GLD leading (with $k = 2 = D$ in Table 1, or in other words, the exactly covariance of SGD noise), GLD Hessian ($k = 2$) and GLD 1st eigven($H$). And we adjust $\sigma_t$ in each dynamics to force their noise to share the same expected squared norm as defined in Eq. (6). Figure 1(a) shows the trajectories of the dynamics escaping from the sharp minimum $(1, 1)$ towards the flat one $(-1, -1)$, while Figure 1(b) presents the success rate of escaping for each dynamic during 100 repeated experiments.

As shown in Figure 1, GLD 1st eigvec($H$) achieves the highest success rate, indicating the fastest escaping speed from the sharp minimum. The dynamics with anisotropic noise aligned with Hessian well, including GLD 1st eigvec($H$), GLD Hessian and GLD leading, greatly outperform GD, GLD const with isotropic noise, and GLD diag with noise poorly aligned with Hessian. These experiments are consistent with our theoretical analysis on Ornstein-Uhlenbeck process shown Proposition 2 and 3, demonstrating the benefits of anisotropic noise for escaping from sharp minima.

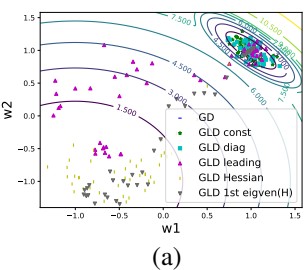 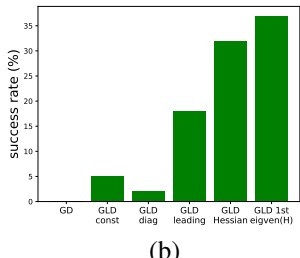 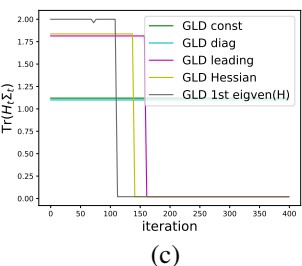

(a)                                  (b)                                  (c)

Figure 1: 2-D toy example. Compared dynamics are defined in Table 1, $k = 2$, $\sigma_t^2$ is tuned to keep noise of all dynamics sharing same expected squared norm, 0.01. All dynamics are run by 500 iterations with learning rate 0.005. **(a)** The trajectory of each compared dynamics for escaping from the sharp minimum in one run. **(b)** Success rate of arriving the flat solution in 100 repeated runs. **(c)** $\mathrm{Tr}(H_t \Sigma_t)$ of compared dynamics in one run.

## 5.2 ONE HIDDEN LAYER NEURAL NETWORK WITH FIXED OUTPUT LAYER PARAMETERS

We empirically show that in one hidden layer neural network with fixed output layer parameters, the anisotropic noise induced by SGD indeed helps escape from sharp minima more efficiently than isotropic noise. Three networks are trained to binary classify $1,000$ linearly separable two-dimensional points. The number of hidden nodes for each network varies in $\{20, 200, 2000\}$. We plot the empirical indicator $\mathrm{Tr}\left(H\Sigma\right)$ in Figure 2. We can easily observe that as the increase of the number of hidden nodes, the ratio $\frac{\mathrm{Tr}(H\Sigma)}{\mathrm{Tr}(H\bar{\Sigma})}$ is enlarged significantly, which is consistent with the Eq. (13) described in Proposition 3.

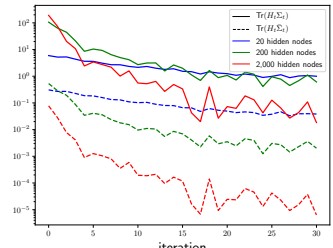

Figure 2: One hidden layer neural networks with fixed output layer parameters. The solid and the dotted lines represent the value of $\mathrm{Tr}(H\Sigma)$ and $\mathrm{Tr}(H\bar{\Sigma})$, respectively. The number of hidden nodes varies in $\{20, 200, 2000\}$, the results of which are denoted in different colors.

## 5.3 PRACTICAL DATASETS

In this part, we conduct a series of experiments in real deep learning scenarios to demonstrate the behavior of SGD noise and its implicit regularization effects. We construct a noisy training set based on FashionMNIST dataset[1]. Concretely, the training set consist of 1000 images with correct labels, and another 200 images with random labels. All the test data are with clean labels. A small LeNet-like network is utilized such that the spectrum decomposition over

---

[1]https://github.com/zalandoresearch/fashion-mnist

gradient covariance matrix and Hessian matrix are computationally feasible. The network consists of two convolutional layers and two fully-connected layers, with $11,330$ parameters in total.

We firstly run the standard gradient decent for 3000 iterations to arrive at the parameters $\theta_{GD}^*$ near the global minima with near zero training loss and $100\%$ training accuracy, which are typically sharp minima that generalize poorly (Neyshabur et al., 2017). And then all other compared methods are initialized with $\theta_{GD}^*$ and run for optimization with the same learning rate $\eta_t = 0.07$ and same batch size $m = 20$ (if needed) for fair comparison[2].

**Verification of SGD noise satisfying the conditions in Proposition 3** To see whether the noise of SGD in real deep learning circumstance satisfies the two conditions in Proposition 3, we run SGD optimizer initialized from $\theta_{GD}^*$, i.e. the sharp minima found by GD.

Figure 3(a) shows the first $400$ eigenvalues of Hessian at $\theta_{GD}^*$, from which we see that the 140th eigenvalue has already decayed to about $1\%$ of the first eigenvalue. Note that Hessian $H \in \mathbb{R}^{D \times D}$, $D = 11330$, thus $H$ around $\theta_{GD}^*$ approximately meets the ill-conditioning requirement in Proposition 3. Figure 3(b) shows the projection coefficient estimated by $\hat{a} = \frac{u_1^T \Sigma u_1 \operatorname{Tr} H}{\lambda_1 \operatorname{Tr} \Sigma}$ along the trajectory of SGD. The plot indicates that the projection coefficient is in a descent scale comparing to $D^{2d-1}$, thus satisfying the second condition in Proposition 3. Therefore, Proposition 3 ensures that SGD would escape from minima $\theta_{GD}^*$ faster than GLD in order of $\mathcal{O}(D^{2d-1})$, as shown in Figure 3(c). An interesting observation is that in the later stage of SGD optimization, $\operatorname{Tr}(H\Sigma)$ becomes significantly ($10^7$ times) smaller than in the beginning stage, implying that SGD has already converged to minima being almost impossible to escape from. This phenomenon demonstrates the reasonability to employ $\operatorname{Tr}(H\Sigma)$ as an empirical indicator for escaping efficiency.

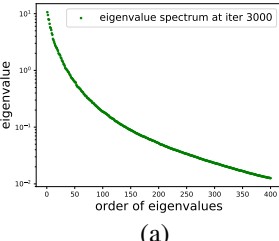 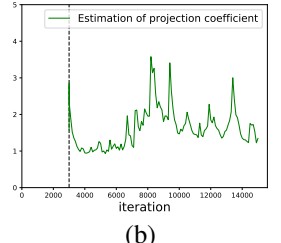 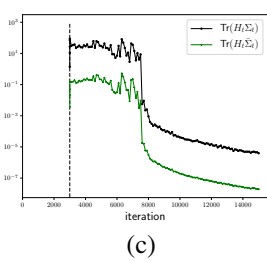

(a)          (b)          (c)

Figure 3: FashionMNIST experiments. **(a)** The first $400$ eigenvalues of Hessian at $\theta_{GD}^*$, the sharp minima found by GD after 3000 iterations. **(b)** The projection coefficient estimation $\hat{a} = \frac{u_1^T \Sigma u_1 \operatorname{Tr} H}{\lambda_1 \operatorname{Tr} \Sigma}$, as shown in Proposition 3. **(c)**$\operatorname{Tr}(H_t \Sigma_t)$ versus $\operatorname{Tr}(H_t \bar{\Sigma}_t)$ during SGD optimization initialized from $\theta_{GD}^*$, $\bar{\Sigma}_t = \frac{\operatorname{Tr} \Sigma_t}{D} I$ denotes the isotropic noise with same expected squared norm as SGD noise.

**Behaviors of different dynamics escaping from minima and its generalization effects** To compare the different dynamics on escaping behaviors and generalization performance, we run dynamics initialized from the sharp minima $\theta_{GD}^*$ found by GD. The settings for each compared method are as follows. The hyperparameter $\sigma^2$ for GLD const has already been tuned as optimal ($\sigma = 0.001$) by grid search. For GLD leading, we set $k = 20$ for comprising the computational cost and approximation accuracy. As for GLD Hessian, to reduce the expensive evaluation of such a huge Hessian in each iteration, we set $k = 20$ and update the Hessian every 10 iterations. We adjust $\sigma_t$ in GLD dynamic, GLD Hessian and GLD 1st eigvec($H$) to guarantee that they share the same expected squred noise norm defined in Eq. (6) as that of SGD. And we measure the expected sharpness of different minima as $\mathbb{E}_{\nu \sim \mathcal{N}(0, \delta^2 I)} \left[ L(\theta + \nu) \right] - L(\theta)$, as defined in ((Neyshabur et al., 2017), Eq.(7)). The results are shown in Figure 4.

As shown in Figure 4, SGD, GLD 1st eigvec($H$), GLD leading and GLD Hessian successfully escape from the sharp minima found by GD, while GLD, GLD dynamic and GLD diag are trapped in the minima. This demonstrates that the methods with anisotropic noise "aligned" with loss curvature can help to find flatter minima that generalize well.

We also provide experiments on standard CIFAR-10 with VGG11 in Appendix.

---

[2]In fact, in our experiment, we test the equally spacing learning rates in the range $[0.01, 0.1]$, and the final results are consistent with each other.

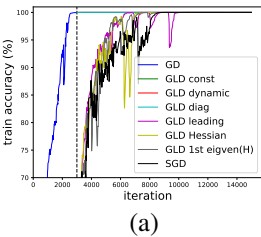 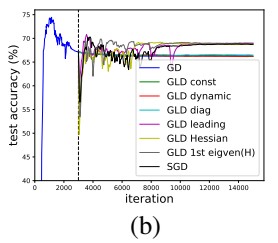 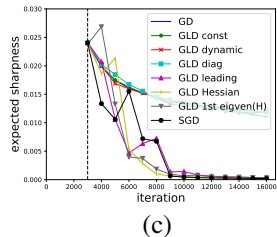

(a)  (b)  (c)

Figure 4: FashionMNIST experiments. Compared dynamics are initialized at $\theta_{GD}^*$ found by GD, marked by the vertical dashed line in iteration 3000. The learning rate is same for all the compared methods, $\eta_t = 0.07$, and batch size $m = 20$. **(a)** Training accuracy versus iteration. **(b)** Test accuracy versus iteration. **(c)** Expected sharpness versus iteration. Expected sharpness is measured as $\mathbb{E}_{\nu \sim \mathcal{N}(0, \delta^2 I)} \left[ L(\theta + \nu) \right] - L(\theta)$, and $\delta = 0.01$, the expectation is computed by average on 1000 times sampling.

## 6 CONCLUSION

We theoretically investigate a general optimization dynamics with unbiased noise, which unifies various existing optimization methods, including SGD. We provide some novel results on the behaviors of escaping from minima and its regularization effects. A novel indicator is derived for characterizing the escaping efficiency. Based on this indicator, two conditions are constructed for showing what type of noise structure is superior to isotropic noise in term of escaping. We then analyze the noise structure of SGD in deep learning and find that it indeed satisfies the two conditions, thus explaining the widely know observation that SGD can escape from sharp minima efficiently toward flat minina that generalize well. Various experimental evidence supports our arguments on the behavior of SGD and its effects on generalization. Our study also shows that isotropic noise helps little for escaping from sharp minima, due to the highly anisotropic nature of landscape. This indicates that it is not sufficient to analyze SGD by treating it as an isotropic diffusion over landscape (Zhang et al., 2017; Mou et al., 2017). A better understanding of this out-of-equilibrium behavior (Chaudhari & Soatto, 2017) is on demand.

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

## A PROOFS OF PROPOSITIONS IN MAIN PAPER

### A.1 PROOF OF PROPOSITION 1

*Proof.* The "mild smoothness assumptions" refers that $L_t = L(\theta_t) \in C^2$. Then the Ito's lemma holds (Øksendal, 2003).

And by Ito's lemma, the SDE of $L_t$ is

$$
\begin{aligned}
\mathrm{d}L_t &= \left( -\nabla L^T \nabla L + \frac{1}{2} \mathrm{Tr}\left( \Sigma_t^{\frac{1}{2}} H_t \Sigma_t^{\frac{1}{2}} \right) \right) \mathrm{d}t + \nabla L^T \Sigma_t^{\frac{1}{2}} \, \mathrm{d}W_t \\
&= \left( -\nabla L^T \nabla L + \frac{1}{2} \mathrm{Tr}\left( H_t \Sigma_t \right) \right) \mathrm{d}t + \nabla L^T \Sigma_t^{\frac{1}{2}} \, \mathrm{d}W_t.
\end{aligned}
$$

Taking expectation with respect to the distribution of $\theta_t$,

$$\mathrm{d}\mathbb{E}L_t = \mathbb{E}\left(-\nabla L^T \nabla L + \frac{1}{2}\operatorname{Tr}(H_t \Sigma_t)\right)\mathrm{d}t, \tag{16}$$

for the expectation of Brownian motion is zero. Thus the solution of $\mathbb{E}Y_t$ is,

$$\mathbb{E}L_t = L_0 - \int_0^t \mathbb{E}\left(\nabla L^T \nabla L\right) + \int_0^t \frac{1}{2}\mathbb{E}\operatorname{Tr}(H_t \Sigma_t)\,\mathrm{d}t.$$

$\square$

## A.2 PROOF OF PROPOSITION 2

*Proof.* Without losing generality, we assume that $L_0 = 0$.

For multivariate Ornstein-Uhlenbeck process, when $\theta_0 = 0$ is an constant, $\theta_t$ follows a multivariate Gaussian distribution (Øksendal, 2003).

Consider change of variables $\theta \to \phi(\theta, t) = e^{Ht}\theta_t$. Here, for symmetric matrix $A$,

$$e^A := U\operatorname{diag}(e^{\lambda_1}, \dots, e^{\lambda_n})U,$$

where $\lambda_1, \dots, \lambda_n$ and $U$ are the eigenvalues and eigenvector matrix of $A$. Note that with this notation,

$$\frac{\mathrm{d}e^{Ht}}{\mathrm{d}t} = He^{Ht}.$$

Applying Ito's lemma, we have

$$\mathrm{d}\phi(\theta_t, t) = e^{Ht}\Sigma^{\frac{1}{2}}\,\mathrm{d}W_t,$$

which we can integrate form $0$ to $t$ to get

$$\theta_t = 0 + \int_0^t e^{H(s-t)}\Sigma^{\frac{1}{2}}\,\mathrm{d}W_s$$

The expectation of $\theta_t$ is zero. And by Ito's isometry (Øksendal, 2003), the covariance of $\theta_t$ is,

$$
\begin{aligned}
\mathbb{E}\theta_t\theta_t^T &= \mathbb{E}\left[\int_0^t e^{H(s-t)}\Sigma^{\frac{1}{2}}\,\mathrm{d}W_s\left(\int_0^t e^{H(r-t)}\Sigma^{\frac{1}{2}}\,\mathrm{d}W_r\right)^T\right] \\
&= \mathbb{E}\left[\int_0^t e^{H(s-t)}\Sigma^{\frac{1}{2}}\Sigma^{\frac{1}{2}}e^{H(s-t)}\,\mathrm{d}s\right] \\
&= \mathbb{E}\left[\int_0^t e^{H(s-t)}\Sigma e^{H(s-t)}\,\mathrm{d}s\right] \\
&= \int_0^t e^{H(s-t)}\Sigma e^{H(s-t)}\,\mathrm{d}s. \quad \text{(for } H \text{ and } \Sigma \text{ are both constant.)}
\end{aligned}
$$

Thus,

$$
\begin{aligned}
\mathbb{E}L(\theta_t) &= \frac{1}{2}\mathbb{E}\operatorname{Tr}\left(\theta_t^T H \theta_t\right) \\
&= \frac{1}{2}\operatorname{Tr}\left(H\mathbb{E}\theta_t\theta_t^T\right) \\
&= \frac{1}{2}\int_0^t \operatorname{Tr}\left(He^{H(s-t)}\Sigma e^{H(s-t)}\right)\mathrm{d}s \\
&= \frac{1}{2}\int_0^t \operatorname{Tr}\left(e^{H(s-t)}H\Sigma e^{H(s-t)}\right)\mathrm{d}s \quad \text{(for } H \text{ is symmetric.)} \\
&= \frac{1}{2}\int_0^t \operatorname{Tr}\left(e^{2H(s-t)}H\Sigma\right)\mathrm{d}s \\
&= \frac{1}{2}\operatorname{Tr}\left(\frac{1}{2}H^{-1}\left(I - e^{-2Ht}\right)H\Sigma\right) \\
&= \frac{1}{4}\operatorname{Tr}\left(\left(I - e^{-2Ht}\right)\Sigma\right).
\end{aligned}
$$

The last approximation is by Taylor's expansion. $\qquad\square$

## A.3 PROOF OF PROPOSITION 3

*Proof.* Firstly, $\operatorname{Tr}(H\Sigma)$ has the decomposition as $\operatorname{Tr}(H\Sigma) = \sum_{i=1}^D \lambda_i u_i^T \Sigma u_i$.

Secondly, compute $\operatorname{Tr}(H\Sigma), \operatorname{Tr}(H\bar{\Sigma})$ respectively,

$$
\operatorname{Tr}(H\Sigma) \geq u_1^T \Sigma u_1 \geq a\lambda_1 \frac{\operatorname{Tr}\Sigma}{\operatorname{Tr}H},
$$

$$
\operatorname{Tr}(H\bar{\Sigma}) = \frac{\operatorname{Tr}\Sigma}{D}\operatorname{Tr}H,
$$

and bound their quotient,

$$
\frac{\operatorname{Tr}(H\Sigma)}{\operatorname{Tr}(H\bar{\Sigma})} \geq \frac{a\lambda_1 D}{(\operatorname{Tr}H)^2} \geq \frac{a\lambda_1 D}{\left(k\lambda_1 + (D-k)D^{-d}\lambda_1\right)^2} = \mathcal{O}\left(aD^{2d-1}\right). \tag{17}
$$

The proof is finished. $\qquad\square$

## A.4 PROOF OF PROPOSITION 4

*Proof.* Firstly compute the gradients and Hessian of $\phi$,

$$
\frac{\partial\phi}{\partial f} = \frac{e^f}{1+e^f} - y = \begin{cases} \frac{e^f}{1+e^f} > 0 & y = 0, \\ -\frac{1}{1+e^f} < 0 & y = 1. \end{cases}
$$

$$
\frac{\partial^2\phi}{\partial f^2} = \frac{e^f}{(1+e^f)^2}.
$$

And note the Gauss-Newton decomposition for functions with the form of $L = \phi \circ f$,

$$
\begin{aligned}
H &= \mathbb{E}_{(x,y)}\frac{\partial^2 \ell((x,y);\theta)}{\partial\theta^2} \\
&= \mathbb{E}_{(x,y)}\frac{\partial^2\phi}{\partial f^2}\frac{\partial f}{\partial\theta}\frac{\partial f^T}{\partial\theta} + \mathbb{E}_{(x,y)}\frac{\partial\phi}{\partial f}\frac{\partial^2 f}{\partial\theta^2}.
\end{aligned}
$$

Since the output layer parameters for $f$ is fixed and the activation functions are piece-wise linear, $f(x;\theta)$ is a piece-wise linear function on its parameters $\theta$. Therefore $\frac{\partial^2 f}{\partial\theta^2} = 0$, a.e., and $H = \mathbb{E}_{(x,y)}\frac{\partial^2\phi}{\partial f^2}\frac{\partial f}{\partial\theta}\frac{\partial f^T}{\partial\theta}$.

It is easy to check that $e^{-C}\left(\frac{\partial\phi}{\partial f}\right)^2 \leq \frac{\partial^2\phi}{\partial f^2} \leq e^C\left(\frac{\partial\phi}{\partial f}\right)^2$. Thus,

$$H = \mathbb{E}_{(x,y)}\frac{\partial^2\phi}{\partial f^2}\frac{\partial f}{\partial\theta}\frac{\partial f^T}{\partial\theta} \preceq \mathbb{E}_{(x,y)}e^C\left(\frac{\partial\phi}{\partial f}\right)^2\frac{\partial f}{\partial\theta}\frac{\partial f^T}{\partial\theta} = \mathbb{E}_{(x,y)}e^C\left(\frac{\partial\phi}{\partial f}\frac{\partial f}{\partial\theta}\right)\left(\frac{\partial\phi}{\partial f}\frac{\partial f}{\partial\theta}\right)^T = e^C F.$$

$$H = \mathbb{E}_{(x,y)}\frac{\partial^2\phi}{\partial f^2}\frac{\partial f}{\partial x}\frac{\partial f^T}{\partial x} \succeq \mathbb{E}_{(x,y)}e^{-C}\left(\frac{\partial\phi}{\partial f}\right)^2\frac{\partial f}{\partial x}\frac{\partial f^T}{\partial x} = \mathbb{E}_{(x,y)}e^{-C}\left(\frac{\partial\phi}{\partial f}\frac{\partial f}{\partial\theta}\right)\left(\frac{\partial\phi}{\partial f}\frac{\partial f}{\partial\theta}\right)^T = e^{-C} F.$$

$\square$

## A.5 PROOF OF PROPOSITION 5

*Proof.* For simplicity, we define $g := \nabla\ell$, $g_0 := \nabla L = \mathbb{E}\nabla\ell$.

The gradient covariance and Fisher has the following relationship,
$$F = \mathbb{E}g\cdot g^T = \mathbb{E}(g_0+\epsilon)(g_0+\epsilon)^T = g_0 g_0^T + \mathbb{E}\epsilon\epsilon^T = g_0 g_0^T + \Sigma.$$

Applying Taylor's expansion to $g_0(\theta)$,
$$g_0(\theta) = g_0(\theta^*) + H(\theta^*)(\theta-\theta^*) + o(\theta-\theta^*) = H(\theta^*)(\theta-\theta^*) + o(\theta-\theta^*).$$

Hence,
$$\left\|g_0(\theta)\right\|_2^2 \leq \|H\|_2^2\|\theta-\theta^*\|_2^2 + o\left(\|\theta-\theta^*\|_2^2\right) = \|H\|_2^2\|\theta-\theta^*\|_2^2 + o\left(\|\theta-\theta^*\|_2^2\right).$$

Therefore, with the condition $\|\theta-\theta^*\|_2 \leq \frac{\sqrt{\delta u^T F u}}{\|H\|_2}$, we have
$$\left\|g_0(\theta)\right\|_2^2 \leq \delta u^T F u + o\left(|\delta|\right).$$

Thus,
$$\frac{u^T\Sigma u}{\operatorname{Tr}\Sigma} = \frac{u^T F u - u^T g_0 g_0^T u}{\operatorname{Tr}F - \operatorname{Tr}(g_0 g_0^T)} \geq \frac{u^T F u - \|g_0\|_2^2}{\operatorname{Tr}F - \|g_0\|_2^2} \geq \frac{u^T F u - \|g_0\|_2^2}{\operatorname{Tr}F}$$

$$= \frac{u^T F u}{\operatorname{Tr}F}\left(1 - \frac{\|g_0\|_2^2}{u^T F u}\right) \geq \frac{u^T F u}{\operatorname{Tr}F}\left(1 - \delta - o\left(|\delta|\right)\right) \geq \frac{u^T F u}{\operatorname{Tr}F}e^{-2\delta},$$

for $\delta$ small enough.

On the other hand, Proposition 4 indicates that $e^{-C}F \preceq H \preceq e^C F$, which means,
$$\forall u, u^T(e^C F - H)u \geq 0$$
$$\text{and } \operatorname{Tr}(H - e^{-C}F) \geq 0.$$

Thus $\frac{u^T F u}{\operatorname{Tr}F} \geq \frac{u^T(e^{-C}H)u}{\operatorname{Tr}(e^C H)}$.

Therefore, for $\lambda$, $u$ being a positive eigenvalue and the corresponding unit eigenvector of $H$, we have
$$\frac{u^T F u}{\operatorname{Tr}F} \geq e^{-2C}\frac{\lambda}{\operatorname{Tr}H}$$
$$\frac{u^T\Sigma u}{\operatorname{Tr}\Sigma} \geq \frac{u^T F u}{\operatorname{Tr}F}e^{-2\delta} \geq e^{-2(C+\delta)}\frac{\lambda}{\operatorname{Tr}H}.$$

$\square$

## B ADDITIONAL EXPERIMENTS

### B.1 DOMINANCE OF NOISE OVER GRADIENT

Figure 5 shows the comparison of gradient mean and the expected norm of noise during training using SGD. The dataset and model are same as the experiments of FashionMNIST in main paper, or as in Section C.2. From Figure 5, we see that in the later stage of SGD optimization, noise indeed dominates gradient.

These experiments are implemented by TensorFlow 1.5.0.

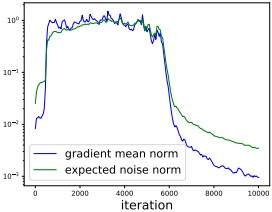

Figure 5: $L_2$ norm of gradient mean, $\|\nabla L(\theta_t)\|$, and the expected norm of noise $\sqrt{\eta_t \mathbb{E}[\epsilon_t^T \epsilon_t]/m}$ during the training using SGD. The dataset and model are same as the experiments of FashionMNIST in main paper, or as in Section C.2

### B.2 THE FIRST 50 ITERATIONS OF FASHIONMNIST EXPERIMENTS IN MAIN PAPER

Figure 6 shows the first 50 iterations of FashionMNIST experiments in main paper. We observe that SGD, GLD 1st eigvec($H$), GLD Hessian and GLD leading successfully escape from the sharp minima found by GD, while GLD diag, GLD dynamic, GLD const and GD do not.

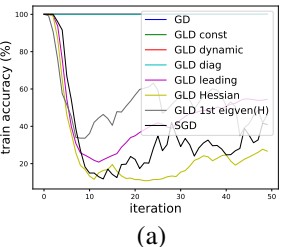
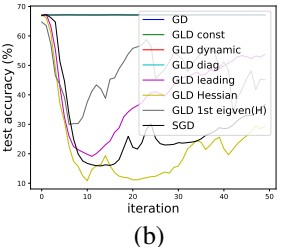

(a)                    (b)

Figure 6: The fisrt 50 iterations of FashionMNIST experiments in main paper. Compared dynamics are initialized at $\theta_{GD}^*$ found by GD. The learning rate is same for all the compared methods, $\eta_t = 0.07$, and batch size $m = 20$. **(a)** Training accuracy versus iteration. **(b)** Test accuracy versus iteration.

These experiments are implemented by TensorFlow 1.5.0.

### B.3 ADDITIONAL EXPERIMENTS ON STANDARD CIFAR-10 AND VGG11

**Dataset**    Standard CIFAR-10 dataset without data augmentation.

**Model**    Standard VGG11 network without any regularizations including dropout, batch normalization, weight decay, etc. The total number of parameters of this network is $9,750,922$.

**Training details**    Learning rates $\eta_t = 0.05$ are fixed for all optimizers, which is tuned for the best generalization performance of GD. The batch size of SGD is $m = 100$. The noise std of GLD constant is $\sigma = 10^{-3}$, which is tuned to best. Due to computational limitation, we only conduct experiments on GD, GLD const, GLD dynamic, GLD diag and SGD.

**Estimation of Sharpness**    The sharpness are estimated by

$$\frac{1}{M}\sum_{j=1}^{M} L(\theta + \nu_j) - L(\theta), \quad \nu_j \sim \mathcal{N}(0, \delta^2 I),$$

with $M = 100$ and $\delta = 0.01$.

**Experiments**    Similar experiments are conducted as in main paper for CIFAR-10 and VGG11, as shown in Figure 7. The observations and conclusions consist with main paper.

These experiments are implemented by PyTorch 0.3.0.

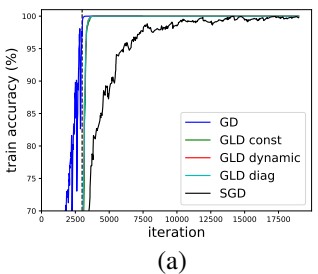 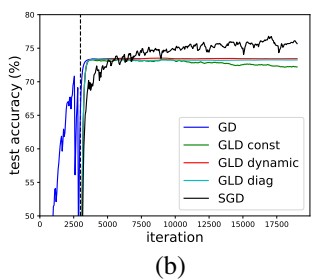 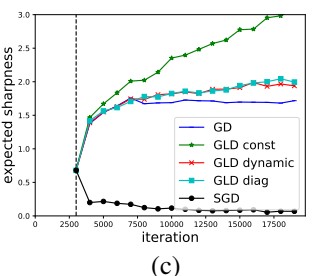

(a)         (b)         (c)

Figure 7: CIFAR-10 experiments. Compared dynamics are initialized at $\theta_{GD}^*$ found by GD, marked by the vertical dashed line in iteration 3000. The learning rate is same for all the compared methods, $\eta_t = 0.05$, and batch size $m = 100$. **(a)** Training accuracy versus iteration. **(b)** Test accuracy versus iteration. **(c)** Expected sharpness versus iteration. Expected sharpness is measured as $\mathbb{E}_{\nu \sim \mathcal{N}(0,\delta^2 I)}\left[L(\theta + \nu)\right] - L(\theta)$, and $\delta = 0.01$, the expectation is computed by average on 100 times sampling. All observations consist with main paper.

## C    DETAILED SETUPS FOR EXPERIMENTS IN MAIN PAPER

### C.1    TWO-DIMENSIONAL TOY EXAMPLE

**Loss Surface**    The loss surface $L(w_1, w_2)$ is constructed by,

$$s_1 = w_1 - 1 - x_1,$$
$$s_2 = w_2 - 1 - x_2,$$
$$\ell(w_1, w_2; x_1, x_2) = \min\{10(s_1 \cos\theta - s_2 \sin\theta)^2$$
$$+ 100(s_1 \cos\theta + s_2 \sin\theta)^2, (w_1 - x_1 + 1)^2 + (w_2 - x_2 + 1)^2\},$$
$$L(w_1, w_2) = \frac{1}{N} \sum_{k=1}^{N} \ell(w_1, w_2; x_1^k, x_2^k),$$

where

$$\theta = \frac{1}{4}\pi,$$
$$N = 100,$$
$$x^k \sim \mathcal{N}(0, \Sigma), \quad \Sigma = \begin{pmatrix} \cos\theta & \sin\theta \\ -\sin\theta & \cos\theta \end{pmatrix}.$$

Note that $\Sigma$ is the inverse of the Hessian of the quadric form generalizeing the sharp minima. And the 3-dimensional plot of the loss surface is shown in Figure 8.

**Hyperparameters**    All learning rates are equal to $0.005$. All dynamics concerned are tuned to share the same expected square norm, $0.01$. The number of iteration during one run is $500$.

These experiments are implemented by PyTorch 0.3.0.

### C.2    FASHIONMNIST WITH CORRUPTED LABELS

**Dataset**    Our training set consists of 1200 examples randomly sampled from original FashionM-NIST training set, and we further specify 200 of them with randomly wrong labels. The test set is same as the original FashionMNIST test set.

**Model**    Network architecture:

$$\text{input} \Rightarrow \text{conv1} \Rightarrow \text{max\_pool} \Rightarrow \text{ReLU} \Rightarrow \text{conv2} \Rightarrow \text{max\_pool}$$
$$\Rightarrow \text{ReLU} \Rightarrow \text{fc1} \Rightarrow \text{ReLU} \Rightarrow \text{fc2} \Rightarrow \text{output}.$$

Both two convolutional layers use $5 \times 5$ kernels with 10 channels and no padding. The number of hidden units between fully connected layers are 50. The total number of parameters of this network are $11,330$.

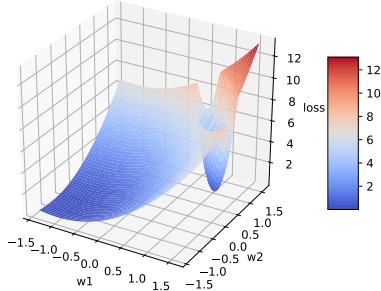

Figure 8: Constructed 2-dimensional surface in main paper.

**Training details**

- **GD**: Learning rate $\eta = 0.1$. We tuned the learning rate (in diffusion stage) in a wide range of $\{0.5, 0.2, 0.15, 0.1, 0.09, 0.08, \ldots, 0.01\}$ and no improvement on generalization.

- **GLD constant**: Learning rate $\eta = 0.07$, noise std $\sigma = 10^{-3}$. We tuned the noise std in range of $\{10^{-1}, 10^{-2}, 10^{-3}, 10^{-4}, 10^{-5}\}$ and no improvement on generalization.

- **GLD dynamic**: Learning rate $\eta = 0.07$.

- **GLD diagnoal**: Learning rate $\eta = 0.07$.

- **GLD leading**: Learning rate $\eta = 0.07$, number of leading eigenvalues $k = 20$, batchsize $m = 20$. We first randomly divide the training set into 60 mini batches containing 20 examples, and then use those minibatches to estimate covariance matrix.

- **GLD Hessian**: Learning rate $\eta = 0.07$, number of leading eigenvalues $= 20$, update frequence $f = 10$. Do to the limit of computational resources, we only update Hessian matrix every 10 iterations. But add Hessian generated noise every iteration. And to the same reason, we simplily set the coeffcent of Hessian noise to $\sqrt{\operatorname{Tr} H / m \operatorname{Tr} \Sigma}$, to avoid extensively tuning of hyperparameter.

- **GLD 1st eigvec($H$)**: Learning rate $\eta = 0.07$, as for GLD Hessian, and we set the coefficient of noise to $\sqrt{\lambda_1 / m \operatorname{Tr} \Sigma}$, where $\lambda_1$ is the first eigenvalue of $H$.

- **SGD**: Learning rate $\eta = 0.07$, batchsize $m = 20$.

**Estimation of Sharpness**    The sharpness are estimated by

$$\frac{1}{M} \sum_{j=1}^{M} L(\theta + \nu_j) - L(\theta), \quad \nu_j \sim \mathcal{N}(0, \delta^2 I),$$

with $M = 1,000$ and $\delta = 0.01$.

These experiments are implemented by TensorFlow 1.5.0.

