# OpenReview forum: "The Anisotropic Noise in Stochastic Gradient Descent: Its Behavior of Escaping from Minima and Regularization Effects"
_ICLR.cc/2019/Conference_

### Official Review · AnonReviewer1 · 2018-10-31
**Interesting but lacks clarity**

**Rating:** 5
**Confidence:** 4

**Review:**

The paper studies the benefit of an anisotropic gradient covariance matrix in SGD optimization for training deep network in terms of escaping sharp minima (which has been discussed to correlate with poor generalization in recent literature).

In order to do so, SGD is studied as a discrete approximation of stochastic differential equation (SDE). To analyze the benefits of anisotropic nature and remove the confounding effect from scale of noise, the scale of noise in the SDE is considered fixed during the analysis. The authors identify the expected loss around a minimum as the efficient of escaping the minimum and show its relation with the hessian and gradient covariance at the minimum. It is then shown that when all the positive eigenvalues of the covariance matrix concentrate along the top eigenvector and this eigenvector is aligned with the top eigenvector of the Hessian of the loss w.r.t. the parameters, SGD is most efficient at escaping sharp minima. These characteristics are analytically shown to hold true for a 1 hidden layer network and experiments are conducted on toy and real datasets to verify the theoretical predictions.

Comments:

I find the main claim of the paper intuitive-- at any particular minimum, if noise in SGD is more aligned with the direction along which loss surface has a large curvature (thus the minimum is sharp along this direction), SGD will escape this minimum more efficiently. On the other hand, isotropic noise will be wasteful because a sample from isotropic noise distribution may point along flat directions of the loss even though there may exist other directions along which the loss curvature is large. However, I have several concerns which I find difficult to point out because *many equations are not numbered*.

1. In proposition 2, it is assumed under the argument of no loss of generality that both the loss at the minimum L_0=0 and the corresponding theta_0 =0. Can the authors clarify how both can be simultaneously true without any loss of generality?
2. A number of steps in proposition 2 are missing which makes it difficult to verify. When applying Ito's lemma and taking the integral from 0 to t, it is not mentioned that both sides are also multiplied with the inverse of exp(Ht).
3. In proposition 2, when computing E[L(theta_t)] on page 12, the equalities after line 3 are not clear how they are derived. Please clarify or update the proof with sufficient details.
4. It is mentioned below proposition 2 that the maximum of Tr(H. Sigma) under constraint (6) is achieved when Sigma* = Tr(Sigma). lambda_1 u1.u1^T, where lambda_1 is the top eigenvalue of H. How is lambda_1 a factor in Sigma*? I think Sigma* should be Tr(Sigma). u1.u1^T because this way the sum of eigenvalues of Sigma remains unchanged which is what constraint (6) states.
5. The proof of proposition 5 is highly unclear.Where did the inequality ||g_0(theta)||^2 <= delta.u^TFu + o(|delta|) come from? Also, the inequality right below it involves the assumption that u^Tg_0 g_0u <= ||g_0||^2 and no justification has been provided behind this assumption.


Regarding experiments, the toy experiment in section 5.1 is interesting, but it is not mentioned what network architecture is used in this experiment. I found the experiments in section 5.3 and specifically Fig 4 and Fig 7 insightful. I do have a concern regarding this experiment though. In the experiment on FashionMNIST in Fig 4, it can be seen that both SGD and GLD 1st eigvec escapes sharp minimum, and this is coherrent with the theory. However, for the experiment on CIFAR-10 in Fig 7, experiment with GLD 1st eigvec is missing. Can the authors show the result for GLD 1st eigvec on CIFAR-10? I think it is an important verification of the theory and CIFAR-10 is a more realistic dataset compared with FashionMNIST.

A few minor points:

1. In the last paragraph of page 3, it is mentioned that the probability of escaping can be controlled by the expected loss around minimum due to Markov's inequality. This statement is inaccurate. A large expected loss upper bounds the escaping probability, it does not control it.
2. Section 4 is titled "The anisotropic noise of SGD in deep networks", but the sections analyses a 1 hidden layes network. This seems inappropriate.
3. In the conclusion section, it is mentioned that the theory in the paper unifies various existing optimization mentods. Please clarify.

Overall, I found the argument of the paper somewhat interesting but I am not fully convinced because of the concerns mentioned above.

---

### Official Review · AnonReviewer3 · 2018-11-02
**A paper analyzing effect of anisotropic noise on SGD dynamics**

**Rating:** 6
**Confidence:** 3

**Review:**

The authors studied the effect of the anisotropic noise of SGD on the algorithm’s ability to escape from local optima. To this end, the authors depart from the established approximation of SGD in the vicinity of an optimum as a continuous-time Ornstein-Uhlenbeck process. Furthermore, the authors argue that in certain deep learning models, the anisotropic noise indeed leads to a good escaping from local optima.

Proposition 3 (2) seems to assume that the eigenvectors of the noise-covariance of SGD are aligned with the eigenvectors of the Hessian. Did I understand this correctly and is this sufficient? Maybe this is actually not even necessary, since the stationary distribution for the multivariate Ornstein-Uhlenbeck process can always be calculated (Gardiner; Mandt, Hoffman, and Blei 2015–2017)

I think this is a decent contribution.

---

### Official Review · AnonReviewer2 · 2018-11-03
**needs more work**

**Rating:** 4
**Confidence:** 5

**Review:**

This paper studies the effort of anisotropic noise in stochastic optimization algorithms. The goal is to show that SGD escapes from sharp minima due to such noise. The paper provides preliminary empirical results using different kinds of noise to suggest that anisotropic noise is effective for generalization of deep networks.

Detailed comments:

1. I have concerns about the novelty of the paper: It builds heavily upon previous work on modeling SGD as a stochastic differential equation to understand its noise characteristics. The theoretical development of this manuscript is straightforward until simplistic assumptions such as the Ornstein-Uhlenbeck process (which amounts to a local analysis of SGD near a critical point) and a neural network with one hidden layer. Similar results have also been in the the literature before in a number of places, e.g., https://arxiv.org/abs/1704.04289 and references therein.

2. Proposition 4 looks incorrect. If the neural network is non-convex, how can the positive semi-definite Fisher information matrix F sandwich the Hessian which may have strictly negative eigenvalues at places?

3. Section 5 contains toy experiments on a 2D problem, a one layer neural network and a 1000-image subset of the FashionMNIST dataset. It is hard to validate the claims of the paper using these experiments, they need to be more thorough. The Appendix contains highly preliminary experiments on CIFAR-10 using VGG-11.

4. A rigorous theoretical understanding of SGD with isotropic noise or convergence properties of Lagevin dynamics has been developed in the literature previously, it’d be beneficial to analyze SGD with anisotropic noise in a similar vein.

---

### Meta-Review · Area_Chair1 · 2018-12-13

**Confidence:** 5
**Recommendation:** Reject

**Metareview:**

The reviewers point our concerns regarding paper's novelty, theoretical soundness, and empirical strength. The authors provided to clarifications to the reviewers.